# Bovine C-X-C Motif Chemokine Ligand 14 Expression Is Regulated by Alternative Polyadenylation and MicroRNAs

**DOI:** 10.3390/ani13193075

**Published:** 2023-09-30

**Authors:** Wei Zhao, Xueyan Liu, Chengping Li, Xuyong Qin, Shizhong Ren, Shujun Cao, Guoli Zhou

**Affiliations:** College of Life Science, Liaocheng University, Liaocheng 252000, China; 2110150210@stu.lcu.edu.cn (W.Z.); 17865154828@163.com (X.L.);

**Keywords:** alternative polyadenylation, preadipocytes proliferation, CXCL14, microRNAs

## Abstract

**Simple Summary:**

The C-X-C motif chemokine ligand 14 (CXCL14) is a chemokine family member that is involved in various cellular responses. However, the post-transcriptional events regulating its expression are poorly understood. In the present study, we identified two 3′ untranslated region (3′ UTR) isoforms of bovine CXCL14 due to alternative polyadenylation (APA) and found that the expression level of the short isoform was significantly higher than that of the long isoform. Further analyses revealed that miR-17-5p, miR-150, and miR-217 reduces expression of the long isoform, indicating that short isoform of the CXCL14 escapes to miRNA-mediated suppression due to APA. Finally, we found that the short APA isoform of CXCL14 promoted preadipocyte proliferation more efficiently than the long one. In conclusion, the results indicate that CXCL14 is post-transcriptionally regulated through APA and microRNAs.

**Abstract:**

Alternative polyadenylation (APA), including APA that occurs only in the 3′ UTR (3′ UTR-APA), is an important post-transcriptional regulatory mechanism that leads to distinct 3′ UTRs for some genes, increasing the complexity of the transcriptome. The post-transcriptional events regulating the expression of bovine, the C-X-C motif chemokine ligand 14 (CXCL14) gene, remains largely unknown. Here, we find that the bovine CXCL14 gene produces two different lengths of mRNA isoforms due to 3′ UTR-APA, and the short and long 3′ UTR is 126 bp and 1155 bp, respectively. We found that the expression level of the short isoform was significantly higher than that of the long isoform by luciferase assays and overexpression of different CXCL14 3′ UTR-APA isoforms. Moreover, using luciferase assay and site-directed mutagenesis experiments, the results showed that the long CXCL14 3′ UTR-APA isoform is downregulated by miR-17-5p, miR-150, and miR-217. However, because the short isoform lacks the true target of miR-17-5p, miR-150, and miR-217 in its 3′ UTR and thus escapes the inhibitory effect of these microRNAs, its expression level is significantly higher than that of the long isoform. Finally, we demonstrate that the short CXCL14 3′ UTR-APA isoform promotes preadipocyte proliferation by cell counting kit 8 (CCK8) assays. Collectively, our results show that the CXCL14 gene is post-transcriptionally regulated through APA and microRNAs.

## 1. Introduction

Beef is an important source of protein, its intramuscular fat (IMF) content affects the tenderness, flavor, and juiciness of meat [1]. Therefore, the improvement of IMF content and beef quality has become an important research direction in recent years. The deposition of IMF is closely related to the proliferation and differentiation of intramuscular preadipocytes [2]. The proliferation process of preadipocytes is strictly regulated by cell cycle regulators such as many cyclin-dependent kinases, cyclin-dependent kinase inhibitors, and other transcription factors [3]. Meanwhile, preadipocyte differentiation is also a complex process regulated by numerous regulatory factors, such as the peroxisome proliferator-activated receptor γ, CCAAT/enhancer-binding protein family, fatty acid-binding protein 4, and lipoprotein lipase [4]. However, in addition to these key genes, many microRNAs (miRNAs) are involved in regulating bovine IMF and beef quality, and their functions and mechanisms need to be further elucidated.

Alternative polyadenylation (APA) is one of the important post-transcriptional regulatory mechanisms. It can be divided into coding region APA and 3′ untranslated region (3′ UTR)-APA, which affects the diversification of mRNA and protein by producing multiple mRNA isoforms of different lengths [5,6,7]. 3′ UTRs of different lengths produced by 3′ UTR-APA affect mRNA stability, translation efficiency, mRNA subcellular or tissue localization, and protein–protein interactions and functions [8,9]. APA switches in most diseases result in the shortening of the 3′ UTR, and this shortening of the 3′ UTR alters the corresponding cis-acting element such as microRNAs (miRNAs) and RNA-binding protein sites, leading to a key consequence of increased mRNA stability and protein expression level [10,11,12,13]. The APA of adipogenesis associated the Mth938 domain-containing (AAMDC) protein gene and heme oxygenase 1 (HO1) gene also play an important role in the differentiation of preadipocytes, respectively [14,15].

The C-X-C motif chemokine ligand 14 (CXCL14) is a small signaling protein produced by all types of cells in the body, with a size of 8-10kDa [16]. CXCL14 is widely expressed in normal tissues, recruits various immune cells to the local environment, and acts as an antibacterial and antitumor factor [17,18]. Growing evidence suggests that CXCL14 is closely related to obesity-induced insulin resistance [19,20]. CXCL14-deficient mice can survive and reproduce with some degree of immune deficiency, but they appear to have a protective effect against obesity-induced insulin resistance [19]. Animal studies have shown that CXCL14 promotes the development of visceral obesity and adipose tissue inflammation, leading to elevated hepatic gluconeogenesis and the development of insulin resistance. Furthermore, Takahashi et al. reported that CXCL14 enhanced insulin-dependent glucose uptake in adipocytes, suggesting improved adipocyte insulin sensitivity [20]. The expression level of CXCL14 is downregulated in obese patients, especially in patients with type 2 diabetes, and is negatively correlated with impaired the glucose/insulin homeostasis index [21]. Overall, the above studies suggest that CXCL14 may play an important role in the biological process of adipocytes.

MiRNAs are single-stranded non-coding RNAs composed of 18–22 nt. They often play an important role in post-transcriptional regulation by combining with the 3′ UTR of the target gene. A miRNA can target the transcripts of multiple genes, and different miRNAs can regulate the transcripts of the same gene [22]. The function of miRNAs is usually to bind to target genes matching the “seed sequence”, inhibit or degrade target gene mRNAs post-transcriptionally, and coordinate normal processes such as cell proliferation, differentiation, and apoptosis [23].

Due to the complex roles of CXCL14 in diverse biological processes, CXCL14 expression must be tightly controlled in post-transcriptional regulation. Despite some progress in the study of CXCL14, the role of its post-transcriptional regulation remains unclear. Therefore, this study sought to verify the existence of 3′ UTR-APA isoforms of the bovine CXCL14 gene by 3′ RACE, and to investigate the post-transcriptional regulatory mechanism of the interaction between different 3′ UTR-APA isoforms of the CXCL14 gene and miRNAs. Meanwhile, we also investigated the effect of different 3′ UTR-APA isoforms of CXCL14 on the proliferation of preadipocytes. This study revealed a novel regulatory mechanism of the bovine CXCL14 gene, which may provide insights into the role of CXCL14 in preadipocyte proliferation.

## 2. Materials and Methods

### 2.1. Cells and Culture Conditions

Human Embryonic Kidney 293T (HEK293T) cells and 3T3-L1 preadipocytes were purchased from Shanghai Cell Bank (Procell, Wuhan, China). The cells were cultured in Dulbecco’s modified Eagle’s medium (DMEM) supplemented with 10% fetal bovine serum (Gibco, Carlsbad, CA, USA), 1% penicillin–streptomycin (Sangon, Shanghai, China), and 2 mM glutamine (Invitrogen, Carlsbad, CA, USA) at 37 °C in a 5% CO_2_ humidified atmosphere.

### 2.2. RNA Isolation and cDNA Synthesis

Total RNA was extracted using the RNeasy Animal RNA Isolation Kit with Spin Column (Beyotime, Shanghai, China). The quality of RNA was tested by 0.1% agarose gels and Nanodrop (Thermo Fisher Scientific, Waltham, MA, USA). The oligo dT was used as the reverse transcription primer of the general qRT-PCR cDNA and 1 μg of RNA was used for reverse transcription using the BeyoRT First Strand cDNA Synthesis Kit (Beyotime, Shanghai, China).

### 2.3. Rapid Amplification of cDNA 3′ Ends

The First Choice™ RLM-RACE Kit (Thermo Fisher Scientific, Waltham, MA, USA) was used to synthesize rapid amplification of cDNA 3′ ends (3′ RACE) cDNA, using M-MLV reverse transcriptase and 1 μg of total RNA was extracted from bovine adipose tissue, according to the manufacturer’s protocol. 3′ RACE PCRs were prepared using the SuperTaq™ Thermostable Taq DNA polymerase following the manufacturer’s protocol. Two forward gene-specific primers were designed according to the RACE Kit guidelines using DNAMAN 6.0 software (Appendix A). The PCR products were cloned and Sanger sequenced according to the conventional method.

### 2.4. Vector Construction

The dual-luciferase reporter gene recombinant was obtained by cloning CXCL14 3′ UTR sequences of different lengths into the restriction sites between *Xho*I and *Not*I of psi-CHECK2 vector (Promega, Madison, WI, USA). The recombinant of psi-CHECK2 vectors containing short and long 3′ UTRs are referred to as 3′ UTR-S and 3′ UTR-L, respectively. The overexpressed recombinants were constructed by using a seamless cloning kit (Beyotime, Shanghai, China) to construct coding sequences (CDS) and CDS with 3′ UTR of different lengths into the downstream of Flag tag of pCMV vector (Clontech Laboratories, Inc., Mountain View, CA, USA). The recombinant of pCMV vectors containing CDS, short, and long 3′ UTR are referred to as CXCL14-CDS, CXCL14-S, and CXCL14-L, respectively. Using the QuickMutation™ Site-Directed Mutagenesis Kit (Beyotime, Shanghai, China), the proximal polyadenylation signal (PAS1) of the 3′ UTR-L and CXCL14-L recombinants was mutated into recombinant 3′ UTR-LM and CXCL14-LM of mutant PAS1 (ATTAAA was mutated to TCCAAA), respectively. In the same way, recombinant 3′ UTR-L was used to construct recombinants with corresponding miRNAs target mutations in the long 3′ UTR, which were called 3′ UTR-L-Mut17, 3′ UTR-L-Mut150, and 3′ UTR-L-Mut217. All primers required for vector construction are shown in Appendix A.

The transfection of HEK293 cells or 3T3-L1 preadipocytes was performed by electroporation. Approximately 3 × 10^6^ cells were mixed with plasmids or miRNA mimics in 500 μL of serum-free medium and then electroporated using Bio-Rad Gene Pulser Xcel (BIO-RAD, Hercules, CA, USA) according to the manufacturer’s protocol. Plasmids were transfected at 1.5 μg per well of a 12-well plate. Co-transfections included 2.0 μg plasmid and miRNA mimics (final concentration 50 nM) or a mimics negative control per well of a 12-well plate. The mimics are summarized in Appendix A.

### 2.5. Dual-Luciferase Reporter Assay

For luciferase reporter assays, HEK293T cells were transfected and harvested after 48 h. According to the manufacturer’s recommendations, the Dual-Luciferase Reporter Assay System (Promega, Madison, WI, USA) was used to measure dual-luciferase activity in cells treated with the Dual-Luciferase Reporter Gene Assay Kit (Beyotime, Shanghai, China). *Renilla* luciferase activity was normalized to the firefly luciferase activity.

### 2.6. Quantitative Real-Time PCR

According to the manufacturer’s instructions, Quantitative Real-Time PCR (qRT-PCR) was performed on the Bio-Rad CFX Maestro system (Bio-Rad, Hercules, CA, USA). The qRT-PCR contained 5 μL of BeyoFast™ SYBR Green qPCR Mix (Beyotime, Shanghai, China), 25 ng of diluted cDNA, and 5 μM of each primer, contributing a total volume of 10 μL. The procedure included initial denaturation at 95 °C for 3 min, followed by 40 cycles for 15 s at 95 °C and 30 s at 60 °C, then at 95 °C for 1 min, and 55 °C for 1 min. GAPDH was selected as loading controls to normalize mRNA expression. The sequences of qRT-PCR primers are summarized in Appendix A.

### 2.7. Western Blot Analysis

The collected cells were lysed with lysis buffer-containing protease inhibitors (Beyotime, Shanghai, China) and placed in a pre-cooled centrifuge, and centrifuged at 13,000× *g* for 20 min. The BCA Protein Assay Kit (Beyotime, Shanghai, China) was then used to quantify the protein concentration. Next, 20 μg of total protein was separated by 12% polyacrylamide gel electrophoresis and transferred to a PVDF membrane by Trans-Blot SD Semi-Dry Electrophoretic Transfer Cell (Bio-Rad, Hercules, CA, USA), and blocked with blocking solution (Beyotime, Shanghai, China) for 1 h at room temperature. After blocking, the membrane was placed at 4 °C to react with the primary antibody overnight and then washed 3 times with TBST and incubated with the secondary antibody for 1 h at room temperature. Finally, it was washed three times with TBST, and the BeyoECL Star Kit (Beyotime, Shanghai, China) was used to visualize the protein bands. According to the manufacturer’s protocol, the image is presented through Amersham Imager 600 (GE Healthcare Bio-Sciences Corp., Piscataway, NJ, USA). The primary antibody information used is as follows: anti-Flag (diluted 1:1000, Abclonal, Wuhan, China), anti-Tubulin (diluted 1:1000, GenScript, Nanjing, China), followed by the horseradish peroxidaseconjugate secondary antibody (diluted 1:10,000, GenScript, Nanjing, China).

### 2.8. Cell Counting Kit 8 (CCK8) Assay

The transfected preadipocytes were seeded into 96-well plates. Then, 24 h after transfection, 10 μL of CCK8 reagent (Beyotime, Beijing, China) was added to each well, and the incubation was continued for 2 h at 37 °C. In addition, the cell proliferation index was assessed at 450 nm using an automated microplate reader (BioTek, Winooski, VT, USA) according to manufacturer’s instructions.

### 2.9. Statistical Analysis

Biostatistical analyses were performed using GraphPad Prism 7.0 (GraphPad Software, San Diego, CA, USA). Comparisons between groups were made using unpaired two-tailed Student’s *t* tests. For comparison of more than two groups with comparable variances, one-way analysis of variance and Bonferroni’s post hoc tests were carried out; *p* < 0.05 was considered statistically significant. All results are presented as the mean ± standard error of the mean (SEM) and were from at least three independent experiments.

## 3. Results

### 3.1. 3′ RACE Identifies Two CXCL14 Alternative 3′ Untranslated Region Variants

To identify the 3′ UTR-APA isoforms of the bovine CXCL14 gene, two gene-specific primers 1 (GSP1) and GSP2 were used to identify the 3′ UTR-APA isoforms of the bovine CXCL14 gene of different lengths by 3′ RACE (Figure 1A). The short 3′ UTR of bovine CXCL14 was amplified with primers GSP1 to obtain a fragment of about 240 bp (Figure 1B, left panel), while the long 3′ UTR of bovine CXCL14 was obtained by GSP2, with a fragment size of about 407bp (Figure 1B, right panel). Sanger sequencing analysis showed that the CXCL14 has two 3′ UTR isoforms of different lengths, the short 3′ UTR is 126 bp (Figure 1C, upper panels), and the long one is 1155 bp (Figure 1C, lower panel). In addition, the short 3′ UTR isoform has a noncanonical polyadenylation signal (PAS) “ATTAAA”, and the long 3′ UTR has a canonical PAS “AATAAA” (Figure 1C).

### 3.2. Short 3′ UTR of CXCL14 Has a Higher Expression Level Than Long 3′ UTR

To explore the effect of different length 3′ UTRs of the CXCL14 gene on the expression of the CXCL14 gene, the long and short 3′ UTRs were constructed into the downstream of psi-CHECK2 vector and named as 3′ UTR-S and 3′ UTR-L, respectively. At the same time, the first signal of 3′ UTR-L ATTAAA was mutated into TCCAAA and named 3′ UTR-LM (Figure 2A). The constructed vectors were transfected into HEK293T cells by electroporation, respectively. The luciferase activity was detected after 48 h. The results of the dual-luciferase assay showed that the luciferase activity of the short 3′ UTR-APA isoform was significantly higher than that of the long 3′ UTR-APA isoforms. However, there was no significant difference between the long isoform of mutant PAS1 and the long isoform of non-mutant PAS1 (Figure 2B). Subsequently, CDS regions of the CXCL14 gene with 3′ UTR of different lengths were cloned into the pCMV vector, respectively (Figure 3A). To investigate the effects of two different 3′ UTR-APA isoforms on the CXCL14 transcription and translation level, the constructed overexpression vector was transfected into HEK293T cells by electroporation. After 48 h, qRT-PCR and Western blot analysis showed that the expression level of the short 3′ UTR-APA isoform was higher than that of the long isoforms. Consistent with the luciferase assay results, there was no significant difference between the long isoform of mutant PAS1 and the long isoform of non-mutated PAS1 at the mRNA and protein levels (Figure 3B,C). These results indicate that the short 3′ UTR-APA isoform of CXCL14 has a higher expression level than the long one.

### 3.3. CXCL14 Is Targeted and Regulated by miR17-5p, miR-150 and miR-217

To further explore the potential mechanism that the long 3′ UTR isoform of CXCL14 is lower in mRNA and protein expression than the short 3′ UTR isoform, we speculate that there may be one or more miRNA-binding sites on the long 3′ UTR of CXCL14 to inhibit its expression. The online software TargetScan 8.0, miRanda v3.3a, RegRNA 2.0, and RNA22 v2 were used to search potential miRNAs predicted to bind the 3′ UTR. After comprehensive analysis of the predicted miRNAs, we selected five putative miRNAs binding to long 3′ UTR, namely, miR-17-5p, miR-150, miR-217, miR-671, and miR-874, for subsequent experimental verification. The binding sites of these miRNAs to CXCL14 are only in its long 3′ UTR, and there is no binding site in its short 3′ UTR (Figure 4A). To verify whether the predicted miRNAs have regulatory effects on the CXCL14 gene, the synthetic miR-17-5p, miR-150, miR-217, miR-671, and miR-874 mimics were co-transfected with 3′ UTR-S or 3′ UTR-L into HEK293T cells, respectively. The results showed that all the 3′ UTR-S + miRNA mimics groups did not significantly reduce the 3′ UTR-S luciferase activity compared with the 3′ UTR-S + mimics NC group, indicating that there is no true binding target for them in the short 3′ UTR (Figure 4B,C). Compared with the 3′ UTR-L + mimics NC group, the luciferase activity in the 3′ UTR-L + miR-671 or miR874 mimics group was not significantly reduced (Figure 4D), but the luciferase activity in the 3′ UTR-L + miR-17-5p, miR150, or miR-217 mimics group was significantly reduced (Figure 4E).

Next, miR-17-5p, miR150, and miR217were co-transferred into HEK293T cells with CXCL14-S or CXCL14-L, respectively. The results showed that miR-671 and miR874 did not significantly down regulate the mRNA and protein expression levels of the CXCL14 gene with short 3′ UTR (Figure 5A,B). However, they significantly down regulated the mRNA and protein expression levels of the CXCL14 gene with long 3′ UTR (Figure 5C,D). Together, these results indicated that miR-17-5p, miR150, and miR217 could bind to the long 3′ UTR of the CXCL14 gene to inhibit its expression.

To further verify the specific effects of miR-17-5p, miR150, miR217 on CXCL14 long 3′ UTR, site-directed mutagenesis was used to mutate the three miRNAs binding sites of the 3′ UTR-L recombinants to form mutations recombinant: 3′ UTR-L-Mut17, 3′ UTR-L-Mut150 and 3′ UTR-L-Mut217 (Figure 6A). These mutant recombinants were co-transfected with NC, miR-17-5p, miR-150 and miR-217 in HEK293T cells, respectively. After 48 h, luciferase activity was detected. The results showed that, compared with the NC group, there was no significant difference in luciferase activity in the group of 3′ UTR-L-Mut17, 3′ UTR-L-Mut150, and 3′ UTR-L-Mut217 (Figure 6B). This result further confirms that miR-17-5p, miR-150, and miR-217 could indeed bind to the long 3′ UTR region of the CXCL14 gene.

### 3.4. The APA Isoform of CXCL14 Gene Differentially Affects the Preadipocytes Proliferation

The formation of fat is not only the process of preadipocyte differentiation into adipocytes, but also involves the process of cell proliferation. To analyze whether CXCL14 plays a role in the stage of preadipocytes proliferation, CXCL14 was overexpressed in 3T3-L1 preadipocytes, and then the expression of cell proliferation marker genes CyclinD1, CyclinE, and Cullin3 was detected by qRT-PCR. The results showed that compared with the control group, overexpression of CXCL14 significantly increased the mRNA expression levels of CyclinD1, Cullin3, and CyclinE (Figure 7A). Subsequently, the CCK-8 assay was used to detect the effects of CXCL14 overexpression on the proliferation of 3T3-L1 preadipocytes at different time points. The results showed that compared with the control group, the overexpression group could promote the proliferation of 3T3-L1 preadipocytes, especially at 48 h (Figure 7B). To further explore the effect of different length 3′ UTR-APA isoforms of the CXCL14 gene on the proliferation of 3T3-L1 preadipocytes, pCMV, CXCL14-S andCXCL14-L were transfected into 3T3-L1 preadipocytes. The expression levels of CyclinD1, Cullin3, and CyclinE were detected by qRT-PCR after 24 h. The results showed that compared with the control group, the expression levels of cell proliferation marker genes CyclinD1, Cullin3, and CyclinE in the CXCL14-S and CXCL14-L groups were significantly upregulated, and the expression levels in the CXCL14-S group were higher than those in the CXCL14-L group (Figure 7C). The proliferation of preadipocytes cultured for different times was detected by CCK-8 assay. The results showed that, compared with the control group, the transfection of CXCL14-S and CXCL14-L groups could promote 3T3-L1 preadipocyte proliferation, but the transfection CXCL14-S group had a more significant promotion effect than the CXCL14-L group, and the promotion effect showed a significant level after 48 h (Figure 7D). These results showed that overexpression of the CXCL14 gene can significantly promote the proliferation of 3T3-L1 preadipocytes, and the short 3′ UTR-APA isoform of CXCL14 has a more significant promoting effect on 3T3-L1 preadipocyte proliferation than the long isoform, indicating that APA of CXCL14 is involved in the regulation of the stage of 3T3-L1 preadipocyte proliferation.

## 4. Discussion

Recent studies have revealed that the isoforms with short 3′ UTRs, generated from APA, exhibit increased stability and translation efficiency by the loss of miRNA-mediated inhibition. For example, Ki-67, AAMDC, and HO1 were post-transcriptionally regulated via APA and miRNAs. APA-mediated shortening of their 3′ UTR contributes to increased mRNA stability and enhanced translational efficiency [13,14,15]. In the present study, the 3′ RACE results showed that CXCL14 long and short 3′ UTR isoforms were expressed in bovine adipose tissues. Then, the effect of long and short 3′ UTR isoforms on protein expression was detected by a dual-luciferase reporter gene assay. The results showed that the short 3′ UTR isoform had higher dual-luciferase activity compared to the long 3′ UTR isoform. To further verify this result, overexpression recombinants coding region + long/short 3′ UTR of the CXCL14 gene were constructed, respectively, and the results displayed by qRT-PCR and Western blot were also consistent with the previous results, which indicated that APA was involved in the regulation of CXCL14 gene expression. It has been reported that the short 3′ UTR isoform produced by APA alters its corresponding cis-acting element, resulting in increased mRNA stability and protein expression as a key result [24].

MiRNAs generally act as negative regulators of gene expression and have important biological significance in the study of post-transcriptional regulatory mechanisms. Five miRNAs targeting CXCL14 were predicted by the miRNAs prediction website: miR-17-5p, miR-150, miR-217, miR-671, and miR-874. The results of the dual-luciferase experiments showed that miR-17-5p, miR-150, and miR-217 could reduce the expression of long 3′ UTR-APA isoforms, while the expression of short 3′ UTR-APA isoforms was not downregulated. After co-transfection of long 3′ UTR with miR-671 and miR-874, the dual-luciferase reporter gene assay showed that the expression of long 3′ UTR was not inhibited. Then, the CXCL14-L overexpression recombinant was co-transfected with miR-17-5p, miR-150, and miR-217. To further verify the targeting relationship between miR-17-5p, miR-150, miR-217, and CXCL14 long 3′ UTR, the qRT-PCR and Western blot results showed that miR-17-5p, miR-150, and miR-217 all inhibited the expression of CXCL14, and the inhibitory effect of miR-217 was significantly stronger than that of miR-17-5p and miR-150. It has been reported that overexpression of miR-17-5p inhibits the differentiation of porcine intramuscular preadipocytes, while knockdown of miR-17-5p promotes the differentiation of porcine intramuscular preadipocytes [25]. Transfection of miR-217 mimics inhibited the proliferation and adipogenic differentiation of 3T3-L1 preadipocytes [26]. The miR-150 can promote the proliferation of bovine preadipocytes and inhibit their differentiation [27]. The miR-150 knockdown mice had lower leptin content, leading to reduced food intake and lower body weight [28]. In addition, for the target binding sites of these three miRNAs located in the long 3′ UTR of the CXCL14 gene, it was found that the miRNAs target sites located at the distal end of the 3′ UTR are more effective than those located at the proximal end [29].

Evidence suggests that chemokines are one of the key factors in the development of pathophysiological processes involved in obesity and insulin resistance [30]. CXCL14 has important research significance in improving the adipose tissue inflammatory state and glucose/insulin homeostasis in obese patients. Studies have reported that the CXCL14 gene is mainly expressed in human adipocytes, and ELISA results also showed that the protein level of CXCL14 in adipocytes was significantly higher than that in preadipocytes, suggesting that it plays an important role in adipogenesis [31]. Likewise, the CXCL14 expression level was elevated in the adipose tissue of obese mice [32]. CXCL14-deficient mice fed a high-fat diet showed impaired fixation of macrophages in white adipose tissue [33], suggesting that CXCL14 is involved in macrophage recruitment in adipose tissue. CXCL14 may recruit macrophages into adipose tissue to produce large amounts of cytokines, chemokines, and growth factors to induce chronic inflammation, thereby promoting the progression of obesity-related diseases such as diabetes and cancer [31]. The regulation of gene expression is often multifactorial because gene expression regulation is a complex and dynamic process, which involves multi-level regulation, including transcriptional, post-transcriptional, translational, and post-translational events. Further investigation is needed to reveal the mechanism by which the bovine CXCL14 gene is strictly regulated, whether in vivo or in vitro. In particular, the mechanism of post-transcriptional regulation needs to study more elements that differentially regulate different 3′ UTR-APA isoforms of CXCL14.

## 5. Conclusions

In conclusion, we identified the presence of APA events in the bovine CXCL14 gene. Our study further confirmed that the short isoform produced by APA has higher expression level and stronger effect of accelerating preadipocyte proliferation than the long isoform due to the inhibitory effect of escape miR-17-5p, miR-150, and miR-217. This research elucidated a novel post-transcriptional regulatory mechanism of the bovine CXCL14 gene by cascading the effects of APA and miRNAs.

## Figures and Tables

**Figure 1 animals-13-03075-f001:**
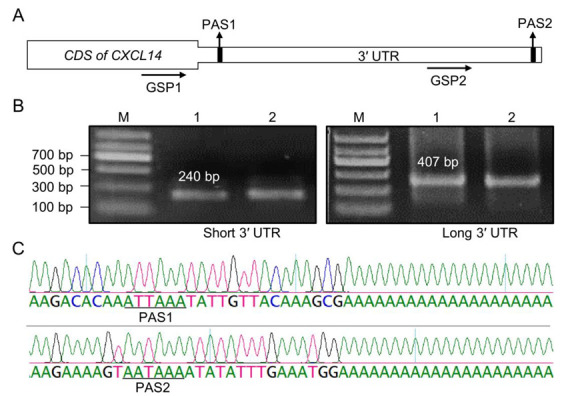
Characterization of APA isoforms of CXCL14 gene. (**A**) Schematic representation of the CXCL14 3′ UTR region. The horizontal arrows indicate the position of the GSP used during 3′ RACE; the position of the proximal PAS (PAS1) and the distal PAS (PAS2) in the 3′ UTR of CXCL14 are shown by vertical arrows. (**B**) Agarose gel electrophoresis profile of the 3′ RACE product of the CXCL14 gene. The left and right panel indicate the short and long APA isoforms of the CXCL14 gene, respectively. M: Molecular weight size marker; Lanes 1 and 2 represent two replicates, respectively. (**C**) The upper and lower panels represent the Sanger sequencing peak maps of the 3′ UTR ends of the short and long APA isoforms of the CXCL14 gene, respectively. PAS1 and PAS2 sequences are underlined.

**Figure 2 animals-13-03075-f002:**
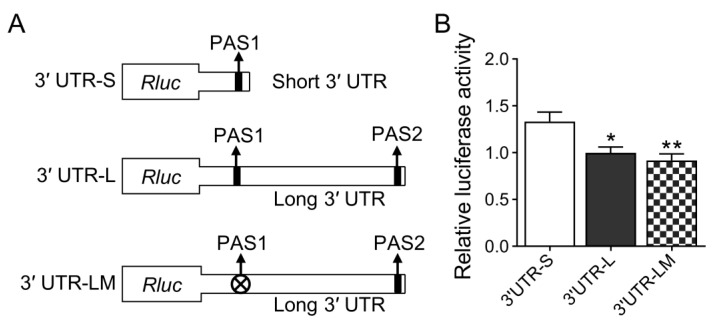
Investigation of the effect of different CXCL14 3′ UTRs on protein expression by luciferase reporter assay. (**A**) Schematic diagram of the CXCL14 3′ UTR-APA isoform’s luciferase reporter recombination. (**B**) Short 3′ UTR of CXCL14 significantly increases the luciferase activity. Renilla luciferase activity was normalized to firefly luciferase activity. All values are represented as means with error bars representing SEM (*n* = 4). *, *p* < 0.05; **, *p* < 0.01.

**Figure 3 animals-13-03075-f003:**
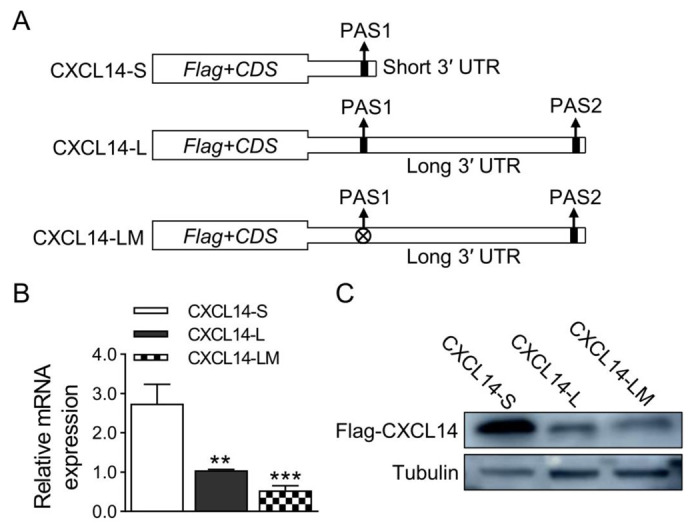
Different effect of CXCL14 with short and long 3′ UTRs on expression level. (**A**) Schematic diagram of the CXCL14 APA isoforms overexpressed recombinants. (**B**) qRT-PCR analysis of isoform abundance in transfected HEK293T cells. GAPDH was used as a loading control (*n* = 4 per group). (**C**) Western blot analysis of transfected HEK293T cells using FLAG antibody. Tubulin was used as a loading control (*n* = 3 per group). All values are represented as means with error bars representing SEM. **, *p* < 0.01; ***, *p* < 0.001.

**Figure 4 animals-13-03075-f004:**
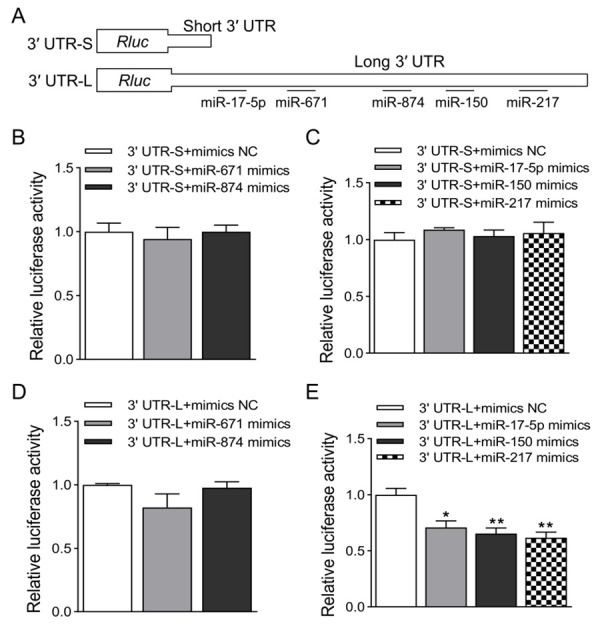
Validation of miRNAs targeting the long 3′ UTR of CXCL14 by dual-luciferase assay. (**A**) Schematic representation of short and long 3′ UTR constructs for luciferase assays, and putative miRNAs that bind only to the long 3′ UTR. (**B**,**C**) Verify miR-671, miR874, miR-17-5p, miR-150, and miR217 targeting CXCL14 short 3′ UTR dual-luciferase assay results. (**D**,**E**) Verify miR-671, miR874, miR-17-5p, miR-150, and miR217 targeting CXCL14 long 3′ UTR dual-luciferase assay results. Renilla luciferase activity was normalized to firefly luciferase activity. All values are represented as means with error bars representing SEM (*n* = 3). *, *p* < 0.05; **, *p* < 0.01.

**Figure 5 animals-13-03075-f005:**
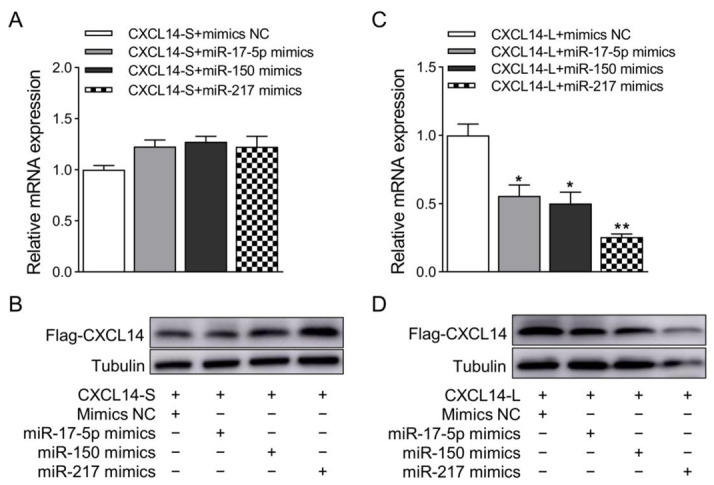
Validation of the interaction between miRNAs and long 3′ UTR CXCL14 isoform by overexpression assays. (**A**) Validation of qRT-PCR results of miR-17-5p, miR150, and miR217 regulating CXCL14 gene APA short 3′ UTR isoform. GAPDH was used as a loading control. (**B**) Western blot detection of the protein expression level of the CXCL14 gene APA short 3′ UTR isoform. Tubulin was used as a loading control. (**C**) Validation of qRT-PCR results of miR-17-5p, miR150, and miR217 regulating CXCL14 gene APA long 3′ UTR isoform. GAPDH was used as a loading control. (**D**) Western blot detection of the protein expression level of the CXCL14 gene APA long 3′ UTR isoform. Tubulin was used as a loading control. All values are represented as means with error bars representing SEM (*n* = 3). *, *p* < 0.05; **, *p* < 0.01.

**Figure 6 animals-13-03075-f006:**
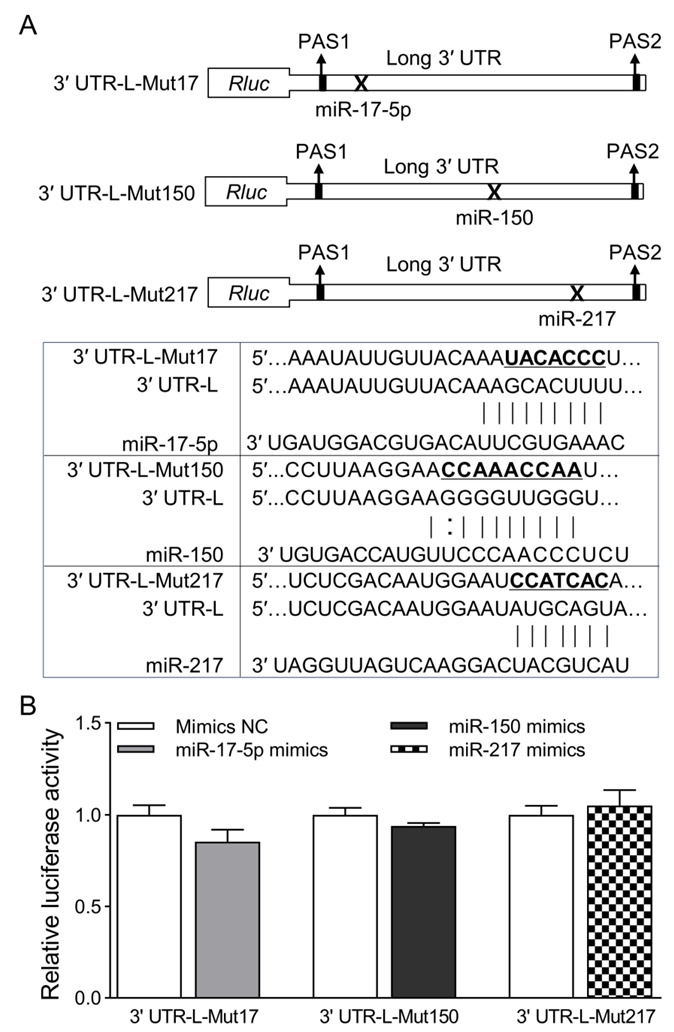
Effect of mutating the miRNAs seed regions on the expression of long 3′ UTR CXCL14 isoform. (**A**) Schematic diagram of long 3′ UTR constructs of miRNA binding site mutation for luciferase assays (upper panel), and mutation site sequences are shown in bold with underline (lower panels). (**B**) Dual-luciferase experimental results of miRNAs and CXCL14 mutant recombinant. Renilla luciferase activity was normalized to firefly luciferase activity. All values are represented as means with error bars representing SEM (*n* = 3).

**Figure 7 animals-13-03075-f007:**
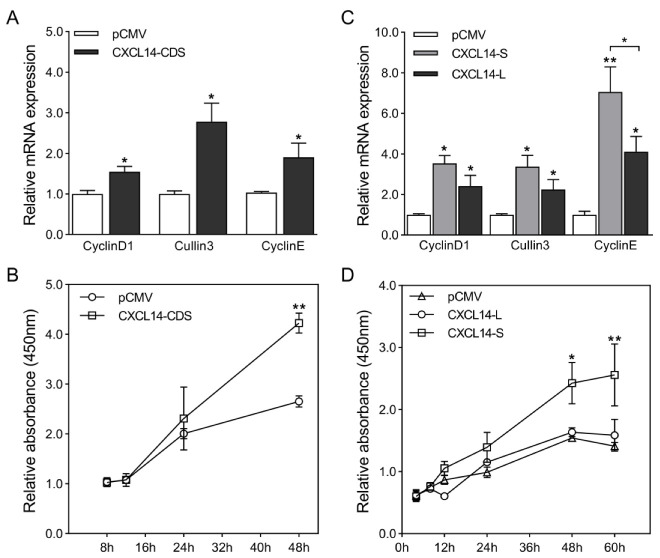
CXCL14 gene and its APA isoforms affect the proliferation of 3T3-L1 preadipocytes. (**A**) After overexpression of CXCL14 CDS, the expression of cell proliferation marker genes CyclinD1, Cullin3, and CyclinE was detected by qRT-PCR. GAPDH was used as a loading control (*n* = 3 per group). (**B**) The CCK-8 assay was used to detect the effect of overexpression of CXCL14 CDS on the proliferation of 3T3-L1 preadipocytes at the designated time points (*n* = 4 per group). (**C**) After overexpression of CXCL14 with long or short 3′ UTR-APA isoforms, the expression of cell proliferation marker genes CyclinD1, Cullin3, and CyclinE was detected by qRT-PCR. GAPDH was used as a loading control (*n* = 3 per group). (**D**) The CCK-8 assay was used to detect the effect of overexpression of CXCL14 with long or short 3′ UTR-APA isoforms on the proliferation of 3T3-L1 preadipocytes at the designated time points (*n* = 3 per group). All values are represented as means with error bars representing SEM. *, *p* < 0.05; **, *p* < 0.01.

## Data Availability

Data are contained within the article.

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
