# Peer review of "Bovine C-X-C Motif Chemokine Ligand 14 Expression Is Regulated by Alternative Polyadenylation and MicroRNAs"

_animals, 2023, doi:10.3390/ani13193075_

Round 1

Reviewer 1 Report

This study reports the identification of 2 mRNA isoforms of the bovine CXCL14 gene. Through in vitro experiments, the study showed that the long isoform is regulated by several miRNAs, and the isoforms differentially affect the preadipocytes proliferation. The authors may consider the following comments.

1)      It is not clear if the authors had any evidence indicating there were 2 isoforms before starting 3’RACE experiment. Was that only based on the presence of PAS1 and PAS2?

2)      It would be ideal to show the presence of both mRNA isoforms in bovine adipose tissue. A northern blot analysis would tell if both isoforms are expressed and the relative levels of them in the cells.

3)      For all experiments, the number of replicates should be indicated in figure legends. For fig 3C, is there a quantitative analysis performed on the western blot data?

4)      Need some more description about how site-directed mutagenesis was performed.

Need some minor corrections.

e.g. line 317: Using CCK-8 to detect 317 cell proliferation at different time points. 

Reviewer 2 Report

Please, check the attached comment file.

I think the quality of English language is adequate.
